# Optomechanical time-domain reflectometry

Gil Bashan[1], Hilel Hagai Diamandi[1], Yosef London[1], Eyal Preter[1] & Avi Zadok[1]

Optical fibres constitute an exceptional sensing platform. However, standard fibres present an inherent sensing challenge: they confine light to an inner core. Consequently, distributed fibre sensors are restricted to the measurement of conditions that prevail within the core. This work presents distributed analysis of media outside unmodified, standard fibre. Measurements are based on stimulated scattering by guided acoustic modes, which allow us to listen where we cannot look. The protocol overcomes a major difficulty: guided acoustic waves induce forward scattering, which cannot be mapped using time-of-flight. The solution relies on mapping the Rayleigh backscatter contributions of two optical tones, which are coupled by the acoustic wave. Analysis is demonstrated over 3 km of fibre with 100 m resolution. Measurements distinguish between air, ethanol and water outside the cladding, and between air and water outside polyimide-coated fibres. The results establish a new sensor configuration: optomechanical time-domain reflectometry, with several potential applications.

[1] Faculty of Engineering and Institute for Nano-Technology and Advanced Materials, Bar-Ilan University, Ramat-Gan 5290002, Israel. These authors contributed equally: Gil Bashan, Hilel Hagai Diamandi, Yosef London. Correspondence and requests for materials should be addressed to A.Z. (email: Avinoam.Zadok@biu.ac.il)

Optical fibre sensors cover hundreds of kilometres, are readily embedded within structures, are immune to electro-magnetic interference, and suitable for hazardous environments[1,2]. Fibres support distributed measurements, in which each segment becomes an independent sensing node of a structural network[3–8]. However, nearly all distributed sensors monitor conditions, such as temperature, strain, sound or vibrations, which affect the core of a standard fibre[3–8]. Known protocols for chemical sensing of surrounding media mandate spatial overlap between light and the substance under test. Hence fibre-based chemical sensors involve non-standard geometries such as photonic-crystal or micro-structured fibres[9–12], make use of reactive materials[13–16] or require significant structural modifications of standard fibres[17–24]. Most realizations support only point sensors. In one notable exception, distributed sensing of chemicals was realized by adding fluorescent dyes to the fibre cladding[25].

We have recently proposed and demonstrated the measurement of the mechanical impedance of substances outside the cladding of an unmodified, standard fibre[26]. The principle is based on guided acoustic waves Brillouin scattering (GAWBS)[27–47], a mechanism in which guided light waves are affected by guided acoustic modes of the fibre. Unlike the optical mode, the transverse profiles of guided acoustic modes extend across the entire cladding cross-section and may probe surrounding media. The mechanical impedance of water and ethanol was measured with 1% precision, even though guided light never reached the test liquids[26]. Several other successful demonstrations quickly followed[48–50]. However, only point-measurements could be realized[26,48–50]. Much greater promise lies in the potential extension of this sensing protocol to distributed measurements. In addition, the previous results were restricted to bare fibres, with the coating completely or partially removed.

Distributed analysis of guided acoustic waves is fundamentally challenging. All known fibre sensor protocols rely on backwards scattering processes, such as Rayleigh, backwards-Brillouin or Raman scattering mechanisms[3–8]. Backscatter events may be directly localized through time-of-flight measurements. Guided acoustic waves, on the other hand, induce forward scattering[27–47]. Here we propose and demonstrate a new sensing concept that works around this difficulty: opto-mechanical time-domain reflectometry (OM-TDR). Analysis is performed over 3 km of standard fibre with 100 m spatial resolution. Measurements successfully distinguish between air, ethanol and water outside the cladding and between acrylate and polyimide coatings. In addition, the analysis also distinguishes between air and water outside a commercially available, polyimide-coated fibre.

## Results

**Principle of operation**. We discuss first the coupling between two optical field components through the stimulation of a guided acoustic wave. Consider the radial guided acoustic modes of the fibre $R_{0,m}$, where $m$ is an integer. Each mode is characterized by a cut-off frequency $\Omega_m$[27,28]. Close to cut-off, the axial acoustic phase velocity approaches infinity[27,28]. Hence for each mode there exists a frequency, very near cut-off, for which the axial phase velocity matches that of the guided optical mode. The acoustic mode may be stimulated by two co-propagating optical field components, which are offset in frequency by $\Omega_m$, through electrostriction.

Let us denote the power levels of two continuous optical fields that co-propagate in the positive $z$ direction as $P_{1,2}(z)$. Their optical frequencies are $\omega_{1,2} = \omega_0 \pm \frac{1}{2}\Omega$, respectively, where $\omega_0$ is a central optical frequency and $\Omega \approx \Omega_m$ a radio-frequency offset.

The two fields are coupled through the stimulation of a guided acoustic wave. One can show that $P_{1,2}(z)$ are governed by the following pair of coupled nonlinear differential equations (Supplementary Note 1):

$$\frac{dP_{1,2}(z)}{dz} = -\alpha P_{1,2}(z) \mp 2\gamma^{(m)}(\Omega, z)P_1(z)P_2(z), \quad (1)$$

Here $\alpha$ denotes the coefficient of linear losses, and $\gamma^{(m)}(\Omega, z)$ is the nonlinear optomechanical coefficient associated with the particular acoustic mode, in units of $W^{-1}km^{-1}$[38,39]. The equations indicate that the stimulation of acoustic waves leads to the amplification of the lower-frequency optical tone, at the expense of the higher-frequency one. They are equivalent to those of forward stimulated Raman scattering, hence the process is also referred to as Raman-like scattering[31]. The nonlinear coefficient $\gamma^{(m)}(\Omega, z)$ is given by[38,39]:

$$\gamma^{(m)}(\Omega, z) = \gamma_0^{(m)}(z) \frac{1}{1 + [\Omega - \Omega_m(z)]^2 / \left[\frac{1}{2}\Gamma_m(z)\right]^2}, \quad (2)$$

where the maximum value $\gamma_0^{(m)}(z)$ is obtained on resonance and $\Gamma_m(z)$ denotes the local modal linewidth (Supplementary Note 1). In standard, single-mode fibres with dual-layer acrylate coating, the process is the most efficient for mode $R_{0,7}$ at $\Omega_7 \sim 2\pi \times 320$ MHz, with $\gamma_0^{(7)} \sim 2.5\ W^{-1}km^{-1}$[40].

We next address the relation between the nonlinear optomechanical coefficient and the properties of surrounding media. The mechanical impedance $Z(z)$ of media outside the cladding modifies the acoustic oscillations. For an infinite medium with impedance that is much lower than that of silica, the effect may be approximated in terms of modifications to the local linewidths $\Gamma_m(z)$[26]. The linewidths consist of two contributions: $\Gamma_m(z) = \Gamma_{m,\text{int}} + \Gamma_{\text{mir}}(z)$. The first is an inherent, mode-dependent linewidth $\Gamma_{m,\text{int}}$, which stems from acoustic dissipation and cladding radius inhomogeneity[32]. Its value for $R_{0,7}$ was previously estimated as $\Gamma_{7,\text{int}} \approx 2\pi \times 0.3$ MHz[26]. The second contribution $\Gamma_{\text{mir}}(z)$ is boundary-related and independent of the choice of $m$. It is determined by the following relation[26]:

$$\left|\frac{Z_f - Z(z)}{Z_f + Z(z)}\right| = \exp\left[-\frac{1}{2}\Gamma_{\text{mir}}(z)t_r\right]. \quad (3)$$

Here $Z_f$ is the mechanical impedance of the fibre itself. For uncoated fibres, the mechanical impedance is that of silica: 13.19 kg mm$^{-2}$ s$^{-1}$. Also in Eq. (3), $t_r$ is the acoustic propagation delay across the fibre diameter[26]. It equals 20.83 ns for uncoated silica fibres with 125 μm diameter. Equation (3) shows that the modal linewidth increases with the mechanical impedance of the outside medium $Z(z)$. The mapping of local optomechanical spectra may therefore provide distributed analysis of media outside the fibre boundary. Such mapping can be extracted from the local power levels of the two optical fields, using Eq. (1):

$$\gamma^{(m)}(\Omega, z) = \frac{1}{4P_1(z)P_2(z)} \left\{ \frac{d[P_2(z) - P_1(z)]}{dz} + \frac{\alpha}{2}[P_2(z) - P_1(z)] \right\}. \quad (4)$$

The OM-TDR measurement protocol is based on Eq. (4). It is illustrated in Fig. 1 (see also Methods section). The amplitudes of the two field components at frequencies $\omega_{1,2} = \omega_0 \pm \frac{1}{2}\Omega$ are jointly modulated by isolated pulses of duration $\tau > 1/\Gamma_m$. Since the local power levels $P_{1,2}(z)$ cannot be measured directly, they are

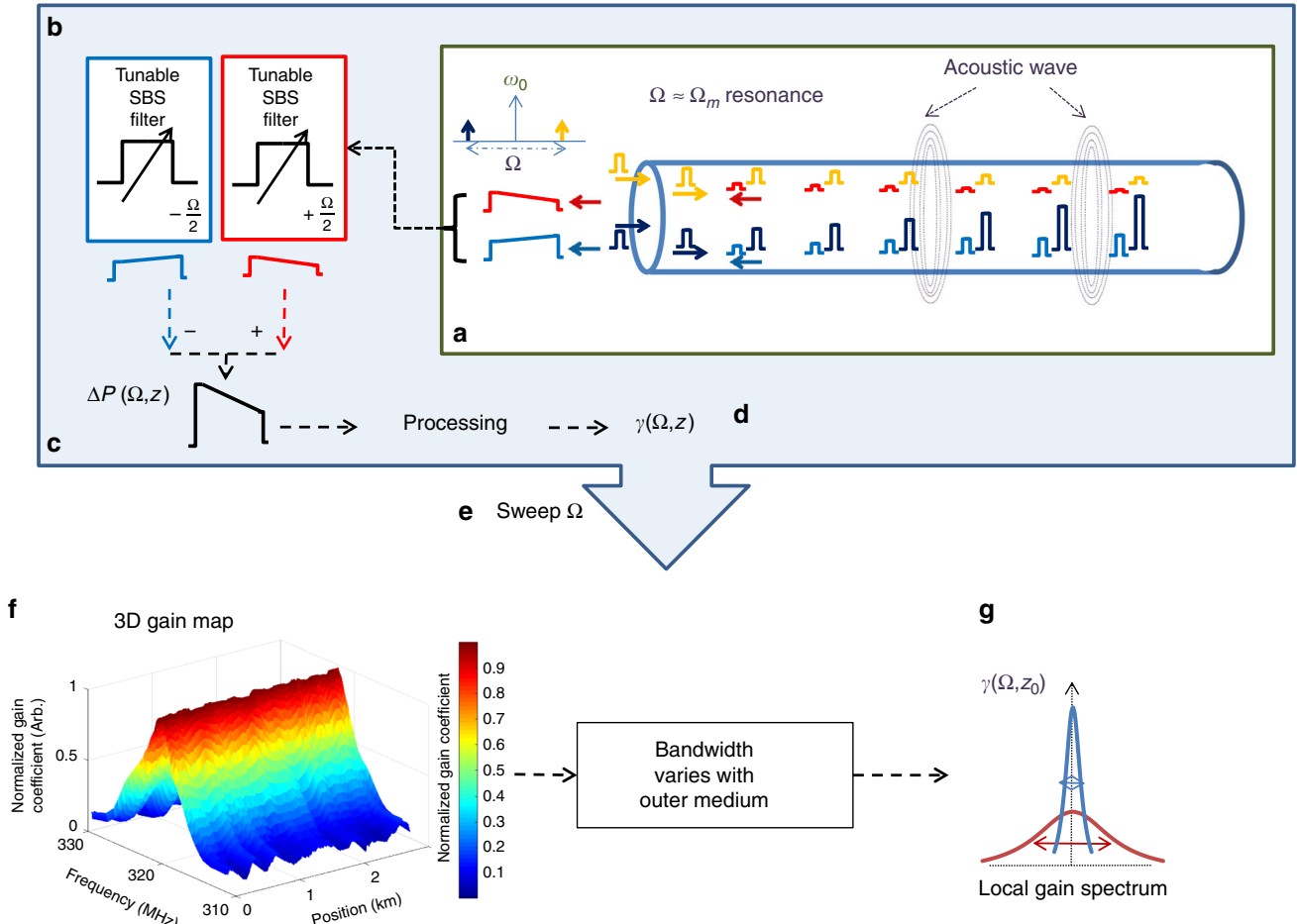

**Fig. 1** The optomechanical time-domain reflectometry principle. Light at the input of a fibre under test consists of two optical tones (marked yellow and dark blue) within a common pulsed envelope (**a**). The two tones are detuned from a central optical frequency $\omega_0$ by radio-frequency offsets $\pm\frac{1}{2}\Omega$. The difference in frequencies $\Omega$ is chosen near the resonance frequency $\Omega_m$ of the stimulation of a particular guided acoustic mode of the fibre $R_{0,m}$. The two pulsed tones are launched into a fibre under test (yellow and dark blue pulses, propagating left to right). The stimulation of acoustic waves along the fibre under test is accompanied by the coupling of optical power between the two tones: the lower-frequency pulses (dark blue) are amplified whereas the higher-frequency ones are attenuated (yellow). The exchange of power is monitored based on the Rayleigh backscatter of the two field components (red and light blue). The two backscatter contributions are separated by narrowband, backwards stimulated Brillouin scattering (SBS) amplifiers (**b**). The waveforms are then detected and sampled for further offline processing. The difference between the two backscatter traces $\Delta P(\Omega, z)$ (**c**) is used to calculate the nonlinear optomechanical coefficient $\gamma^{(m)}(\Omega, z)$, as a function of position $z$, for the particular choice of $\Omega$ (**d**). The measurement sequence is repeated for multiple values of radio-frequency offsets $\Omega$ (**e**), to reconstruct a three-dimensional map of optomechanical coupling as a function of both position and frequency (**f**). A cross-section of that map describes the local spectrum of the optomechanical coefficient $\gamma^{(m)}(\Omega, z_0)$ at location $z_0$ (**g**). The optomechanical coupling spectrum $\gamma^{(m)}(\Omega, z_0)$ varies with the mechanical impedance of the medium outside the fibre at $z_0$. The spectrum is narrow with a sharp peak for an uncoated section of fibre in air (illustrated in blue curve). It becomes broader when the same section is immersed in liquid (red curve)

estimated instead based on the Rayleigh backscatter of the two field components. Multi-wavelength analysis of Rayleigh back-scatter was previously proposed and employed in the distributed monitoring of four-wave mixing processes related to the Kerr effect, and in measurements of local chromatic dispersion[51,52]. A similar approach is followed here towards the mapping of stimulated guided acoustic waves.

The separation between the two backscatter contributions is challenging due to the small offset $\Omega$ between their optical frequencies: only few hundreds of MHz. Individual traces are obtained using narrowband, backwards stimulated Brillouin scattering fibre amplifiers[52], that are tuned to amplify light at either $\omega_1$ or $\omega_2$ (Fig. 1). The two traces are detected and processed offline to obtain the nonlinear coefficient $\gamma^{(m)}(\Omega, z)$ as a function of position, for the particular choice of $\Omega$, using Eq. (4). The procedure is then repeated for multiple choices of $\Omega$, to obtain a

three-dimensional map of the optomechanical nonlinear coefficient as a function of both position and frequency. The spatial resolution is on the order of $\frac{1}{2}v_g\tau$, where $v_g$ is the group velocity of light in the fibre. Access to only one end of the fibre under test is required.

The estimates of $P_{1,2}(z)$ raise an additional challenge: Rayleigh backscatter of coherent light is extremely noisy. Standard optical time-domain reflectometry avoids this difficulty by using broadband, incoherent sources, which are inapplicable to the stimulation of guided acoustic waves. Instead, each OM-TDR trace is acquired multiple times, using different choices of the central optical frequency $\omega_0$, while keeping the frequency offset $\Omega$ fixed. The averaging of collected traces with respect to $\omega_0$ reduces the coherent Rayleigh backscatter noise considerably. However, the need for a large number of repeating measurements at every $\Omega$ is a drawback of

the proposed protocol. Note that while the nonlinear coefficient spectra are constructed using measurements of intensity, the analysis of surrounding media is based on the local linewidths. The spectral estimate is more robust to measurement noise than individual intensity readings.

The OM-TDR principle differs from that of our initial, integrated-sensor demonstration[26], in the following significant respect: Here the acoustic waves are driven by two optical tones, and their counter-effect on the same two tones is being monitored. The process is therefore one of stimulated scattering. In contrast, the monitoring of the acoustic waves in the earlier work relied on the scattering of a separate optical probe wave at a different wavelength[26]. The probe did not affect the acoustic waves, and its scattering may be viewed as a spontaneous process.

**Experimental measurements**. The proposed OM-TDR protocol was demonstrated experimentally. The setup and procedures are described in detail in the Methods section. The duration of incident pulses was 1 µs, corresponding to a spatial resolution of 100 m. Measurements were taken over 2–3 km of standard single-mode fibres with standard, dual-layer acrylate coating of 250 µm diameter. Figure 2 shows pairs of OM-TDR traces of the higher-frequency and lower-frequency optical field components. The traces are averaged with respect to $\omega_0$ as noted above. Dashed lines correspond to a modulation frequency $\Omega = 2\pi \times 340$ MHz that is detuned from the resonance $\Omega_7$ by

several linewidths $\Gamma_7$. The two traces in this case are nearly identical and affected primarily by linear losses. Small-scale differences between the two traces are due to residual coupling through a torsional-radial guided acoustic mode[27,39]. The stimulation of that mode is much less effective than that of the radial mode $R_{0,7}$[39]. Solid lines present a pair of backscatter traces with $\Omega = \Omega_7 = 2\pi \times 321$ MHz (on resonance). The traces indicate that the optical power $P_2(z)$ is amplified along the fibre under test due to optomechanical coupling, whereas $P_1(z)$ is attenuated, as expected. The changes in power are not symmetric: the relative attenuation of $P_1(z)$ exceeds the amplification of $P_2(z)$. The difference is due to residual four-wave mixing effects, which lead to the transfer of power to additional spectral components at frequencies $\omega_0 \pm \frac{3}{2}\Omega$ (see also in the Discussion section below).

Figure 3 shows OM-TDR maps of the nonlinear optomechanical coefficient $\gamma^{(7)}(\Omega, z)$. Data are normalized so that the areas $\int \gamma^{(7)}(\Omega, z)d\Omega$ are the same for all measurements and all

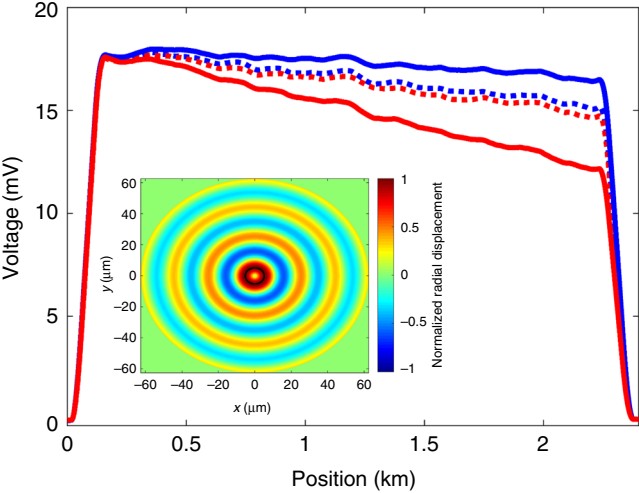

**Fig. 2** Optomechanical time-domain reflectometry traces. The fibre under test was 2.5 km long. Red traces present the Raleigh backscatter contributions of an optical field at frequency $\omega_1 = \omega_0 + \frac{1}{2}\Omega$, where $\omega_0$ is a central optical frequency and $\Omega$ is a radio-frequency offset. The offset is chosen near the resonance frequency $\Omega_7$ of optomechanical coupling due to radial mode $R_{0,7}$. Blue traces present the Rayleigh backscatter of a co-propagating field of frequency $\omega_2 = \omega_0 - \frac{1}{2}\Omega$. Traces were averaged over 1024 discrete choices of $\omega_0$ within a wavelength range of 1558.5–1559.5 nm, with the offset $\Omega$ fixed, in order to reduce coherent contributions to the Rayleigh backscatter. Dashed lines: $\Omega$ detuned from $\Omega_7$ by several linewidths $\Gamma_7$. Little coupling takes place between the two optical fields, and their local power levels are nearly equal along the entire fibre. Both are attenuated due to linear losses. Solid lines: $\Omega$ adjusted to match $\Omega_7$. Attenuation of the lower-frequency optical field is reduced due to optomechanical coupling (solid blue), whereas that of the higher-frequency components is increased (solid red). The inset shows the calculated normalized transverse profile of the material displacement of mode $R_{0,7}$, in a silica fibre with 125 µm diameter, as a function of Cartesian coordinates $x$ and $y$

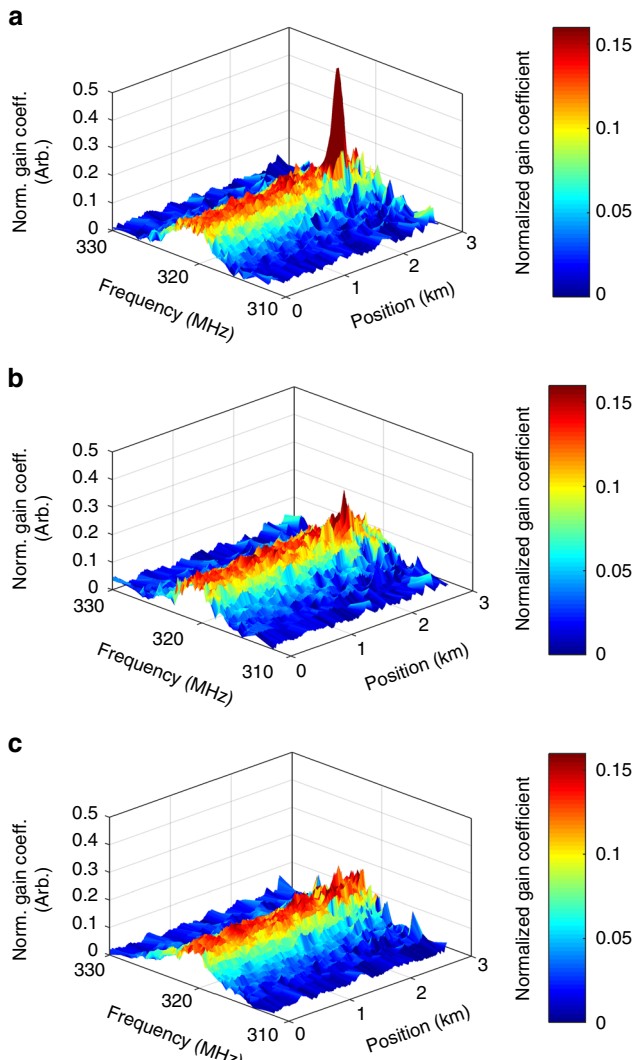

**Fig. 3** Optomechanical time-domain reflectometry maps. The nonlinear coefficients $\gamma^{(7)}(\Omega, z)$ are presented as functions of frequency detuning $\Omega/(2\pi)$ between two co-propagating optical fields and position $z$ along a fibre under test. The nonlinear coefficients are normalized so that $\int \gamma^{(7)}(\Omega, z)d\Omega$ is the same for all measurements and all positions $z$. The dual-layer acrylate coating was removed from a 100 m-long fibre section, located 2.5 km from the input end. The uncoated section was kept exposed in air (**a**), immersed in ethanol (**b**) or immersed in deionized water (**c**)

positions $z$. Narrowband optomechanical coupling is observed in all fibre locations as anticipated. While OM-TDR is applicable to fibres with suitable coating (see also below), measurements were first performed using sections of fibres with the coating removed. A 100 m-long uncoated fibre section at $z = 2.5$ km was kept in air (panel a), or immersed in ethanol (b) or deionized water (c). Figure 4 shows examples of local OM-TDR spectra for a coated fibre section, and for the uncoated section in air, ethanol and water. All curves are well-fitted by Lorentzian line-shapes.

Figure 5a presents the full-widths-at-half maximum of the fitted OM-TDR curves for the three experiments, as a function of position. The fitted resonance frequencies are shown in Fig. 5b. The resonance frequency is $321.15 \pm 0.15$ MHz. The linewidth for coated fibre segments is $5.5 \pm 0.35$ MHz, corresponding to an impedance of $1.9 \pm 0.15$ kg mm$^{-2}$ s$^{-1}$. The measured linewidths for the uncoated section in air, ethanol and water are 1.1 MHz, 2.8 MHz and 4.5 MHz, respectively. The difference between the measured linewidths in water and ethanol is larger than the experimental uncertainty. The linewidths may therefore be used to locally distinguish between the two liquids. The expected linewidths for the uncoated section in air, ethanol and water are 1.1 MHz, 3.3 MHz and 4.5 MHz, respectively. (Note that the linewidths are broadened by the finite duration of incident pulses). The measured linewidths for ethanol and water are in quantitative agreement with predictions. The experimental linewidth for the uncoated fibre in air is broadened further due to the long acoustic lifetime of 1 μs[26], which is comparable to the duration of pulses. The stimulation of the acoustic wave at that section does not fully reach a steady state.

Figure 6a shows an OM-TDR map of a 2.6 km long standard single-mode fibre with polyimide coating of 140 μm diameter. The first 850 m of the fibre under test were immersed in water, whereas the remainder of the fibre was kept in air. Narrowband optomechanical coupling is observed along the entire length of the fibre with a resonance frequency of $369.5 \pm 0.05$ MHz. This value matches the calculated cut-off frequency of radial mode $R_{0,11}$ of the combined silica-polyimide cross-section in air, (see transverse profile in Fig. 6b[53]). Once more, the measured coupling spectra are considerably broader in the fibre section immersed in water: the measured linewidths are $4.6 \pm 0.4$ MHz in water, as opposed to $1.2 \pm 0.1$ MHz in air (panels (c) through (e)). The quantitative analysis of media outside polyimide-coated fibres requires further work. Nevertheless, these preliminary observations illustrate that the OM-TDR principle is applicable to coated fibres as well.

Finally, an OM-TDR map with 3 km range and higher spatial resolution of 50 m is presented in Fig. 7. In this experiment, the duration of pulses was shortened to 500 ns. In addition, the linewidth of the tuneable laser diode used as the source of the two input tones was broadened to 100 MHz, through external phase modulation (see also Methods section). The spectral broadening reduces noise due to coherent Rayleigh backscatter in each trace[54,55]. Therefore, the signal-to-noise ratio (SNR) of the analysis could be improved without increasing the number of averaged traces. In addition, the scanning range of the central optical frequency $\omega_0$ was increased from 1 nm to 2 nm. A 50 m-long section of fibre with 140 μm-diameter polyimide coating was spliced 2 km from the input end of a fibre under test with standard, dual-layer acrylate coating. This section is clearly identified by an optomechanical resonance at 325 MHz frequency, which corresponds to the cut-off of mode $R_{0,10}$ of the polyimide-coated fibre. Mode $R_{0,7}$ of the silica fibre is observed in all ther locations, as before. The results suggest that the spatial resolution of the OM-TDR sensing protocol may be improved further.

## Discussion

The spatial resolution of the distributed analysis is limited by the lifetimes of stimulated guided acoustic waves, and by the SNR of

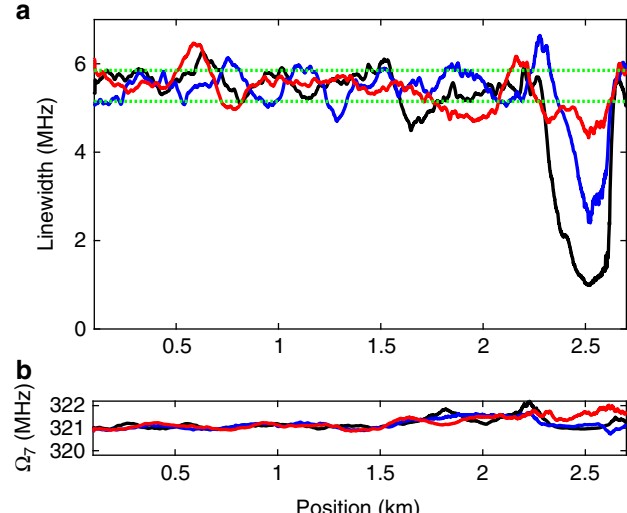

**Fig. 5** Optomechanical time-domain reflectometry measurements.
**a** Measured linewidths $\Gamma_7(z)/2\pi$ as a function of position. **b** Measured resonance frequencies $\Omega_7/(2\pi)$ as a function of position. The standard dual-layer acrylate coating was removed from a 100 m-long section of the fibre under test, located 2.5 km from the input end. Black: uncoated section kept in air. Blue: uncoated section immersed in ethanol. Red: uncoated section immersed in deionized water. The range of experimental uncertainty in the linewidth measurements for coated fibre segments is noted by horizontal lines in (**a**). The uncertainty is estimated based on the standard deviation of the measured local linewidths across all coated sections

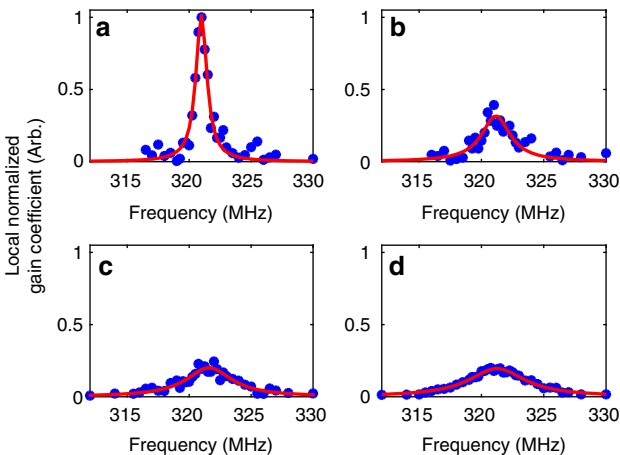

**Fig. 4** Optomechanical time-domain reflectometry spectra. Measured normalized nonlinear coefficients $\gamma^{(7)}(\Omega)$ at specific fibre locations (blue circular signs) are shown alongside fitted Lorentzian line-shapes (red solid traces). The nonlinear coefficients are normalized so that $\int \gamma^{(7)}(\Omega)d\Omega$ is the same for all measurements. A standard, single-mode fibre was under test. **a** A 100 m-long, uncoated fibre section in air, 2.5 km from the input end. **b** Same fibre section immersed in ethanol. **c** Same fibre section immersed in deionized water. **d** A 100 m-long fibre section with standard dual-layer acrylate coating, 2.2 km from the input end. The four curves may be distinguished based on their linewidths. The full-widths at half-maximum of the optomechanical spectra in the four panels are 1.1 MHz, 2.8 MHz, 4.5 MHz and 5.5 MHz, respectively

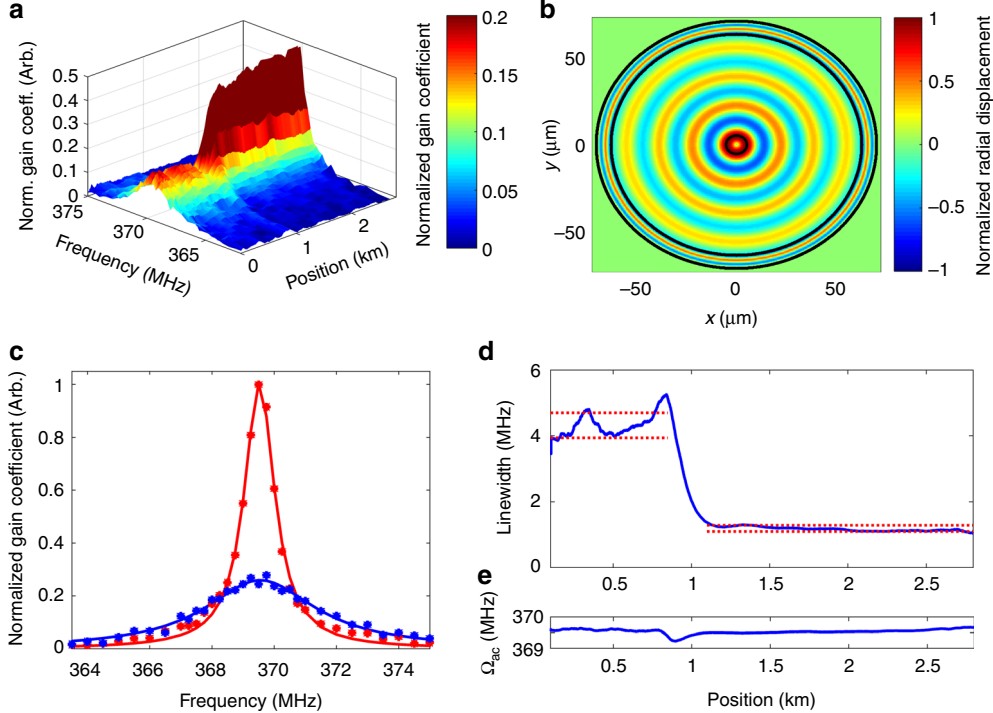

**Fig. 6** Optomechanical time-domain reflectometry of a polyimide-coated fibre. **a** Optomechanical time-domain reflectometry map of the nonlinear coefficient $\gamma(\Omega, z)$ as a function of frequency detuning $\Omega/(2\pi)$ between two co-propagating optical fields and position $z$ along a fibre under test. The nonlinear coefficients are normalized so that $\int \gamma(\Omega, z)d\Omega$ is the same for all positions $z$. The standard single-mode fibre under test is coated with polyimide with outer diameter of 140 μm. The first 850 m of the fibre were immersed in water, and the rest was kept in air. Optomechanical coupling with a resonance frequency of 369.5 MHz due to mode $R_{0,11}$ of the coated fibre is observed throughout its length. The coupling spectrum in air is narrower with a stronger peak. **b** Calculated normalized transverse profile of the material displacement of the radial mode $R_{0,11}$ that is guided by the silica-polyimide structure, as a function of Cartesian coordinates $x$ and $y$. **c** Spectra of optomechanical coupling in fibre sections 750 m from input end (blue, in water), and 2.5 km from the input end (red, in air). Markers present measurement data, and lines show fitted Lorentzian line-shapes. **d** Linewidths $\Gamma_{11}(z)/(2\pi)$ as a function of position. The ranges of experimental uncertainty in the linewidth measurements, for fibre in water and air, are noted by horizontal dotted lines. The uncertainties are estimated by the standard deviation of the measured linewidths in each segment. **e** Resonance frequencies $\Omega_{11}(z)/(2\pi)$ as a function of position

the collected traces. The lifetimes can be as long as 1 μs for fibres in air[26]. The corresponding restriction on resolution can be overcome, for example, by carrying over double-pulse analysis protocols that were developed for Brillouin sensing[56,57]. The measurement SNR currently represents the more stringent restriction. It is affected by the differential analysis of measured traces: $d[P_2(z) - P_1(z)]/dz$, which is susceptible to noise. In similarity with Brillouin sensors, the analysis of the entire gain spectra is more robust to noise than individual measurements of intensity[58]. Nevertheless, better SNR of individual acquisitions would be required for higher resolution[58].

The dominant noise mechanism is due to coherent Rayleigh backscatter. As already demonstrated in Fig. 7 above, this noise mechanism may be partially suppressed through the spectral broadening of the input tones[54,55]. Broadening is based on phase modulation only, hence it does not affect the stimulation of the acoustic modes. The source linewidth should be kept below $\Omega_m$, in order to separate the backscatter traces of the two tones. Future double-pulse analysis protocols would involve pulses that are several μs-long, as opposed to the 500 ns duration used above. When longer pulses are used, the suppression of coherent backscatter through spectral broadening becomes more effective[54,55]. Therefore, the prospects of double-pulse OM-TDR with a spatial resolution of few tens of metres appear promising. In addition, the range of central optical frequencies used in this work has been limited to 125–250 GHz, (the equivalent of 1–2 nm wavelength span), due to equipment constraints. A broader scanning range

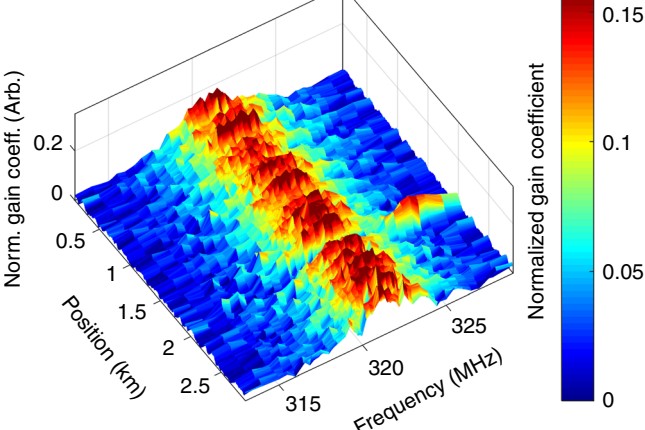

**Fig. 7** Optomechanical time-domain reflectometry with 50 m resolution. Measured nonlinear coefficient $\gamma(\Omega, z)$ as a function of frequency detuning $\Omega/(2\pi)$ between two co-propagating optical fields and position $z$ along a fibre under test. The nonlinear coefficients are normalized so that $\int \gamma(\Omega, z)d\Omega$ is the same for all positions $z$. The fibre under test had standard, dual-layer acrylate coating. Optomechanical coupling due to mode $R_{0,7}$ of the silica fibre is observed throughout its length, with a resonance frequency of 321 MHz. A 50 m-long section of standard single-mode fibre with polyimide coating of 140 μm diameter was spliced 2 km from the input end, and kept in air. An optomechanical resonance at 325 MHz is observed at that location, corresponding to mode $R_{0,10}$ of the coated fibre

may reduce noise levels even further. Measurements outside coated fibres also provide opportunities: the composition and diameter of the coating may be optimized for increased OM-TDR sensitivity, at specific target values of mechanical impedance outside the fibre.

The measurement range is currently restricted by the four-wave mixing build-up of higher-order Stokes and anti-Stokes waves at frequencies $\omega_0 \pm \left(n + \frac{1}{2}\right)\Omega$, where $n$ is an integer[59]. The power levels of four-wave mixing terms at the output end of the fibre were kept at least 6 dB below those of the two primary tones of interest. This requirement restricted the measurement range to 3 km. The OM-TDR protocol may be extended to the monitoring of multiple scattering orders, with proper extension of Eq. (4), to obtain a longer measurement range. The model can also be extended to include Kerr nonlinearity[49]. Although only a modest number of 30 resolution points has been achieved in this work, the limitation is not fundamental. We well expect that both range and spatial resolution will be enhanced in future studies.

In conclusion, three categories of distributed fibre sensors have been known for over 30 years, based on backwards Rayleigh, Raman and Brillouin scattering. All are restricted to the monitoring of conditions that prevail within the core of standard fibre. Herein a fourth class has been proposed and demonstrated: OM-TDR. This new sensor implementation provides distributed analysis of media outside the cladding of standard, unmodified fibre, and even coated fibre. It relies on the mapping of forward scattering by stimulated guided acoustic modes of the entire fibre cross-section. On top of

conceptual novelty, the results may find significant applications in the oil and gas and energy sectors, oceanography, leak detection, structural health monitoring and more.

## Methods

**Stimulation of guided acoustic waves.** A schematic illustration of the OM-TDR experimental setup is shown in Fig. 8. Light from a tuneable laser diode of 100 kHz linewidth, at optical frequency $\omega_0$, is used as the source of all optical fields. In the experiments reported in Figs. 6, 7, the laser diode output is phase-modulated by a pseudo-random binary sequence at a rate of 100 Mbit s$^{-1}$. Modulation is carried out using an external electro-optic phase modulator driven by a pattern generator. The modulation broadens the linewidth of the laser source to the order of the bit rate, and helps reduce noise due to coherent Rayleigh backscatter[54,55].

The laser diode output is then split into two paths. Light in one arm (referred to as the sensor branch) is modulated by a double-sideband electro-optic amplitude modulator, which is driven by a sine wave at a variable radio frequency $\frac{1}{2}\Omega \approx \frac{1}{2}\Omega_m$. The modulator is biased for carrier suppression. Light at the modulator output therefore consists of two field components, at optical frequencies $\omega_{1,2} = \omega_0 \pm \frac{1}{2}\Omega$. The magnitude of the driving voltage is kept below $0.1 V_\pi$ of the modulator, to reduce higher-order modulation harmonics. The reduction of higher-order tones at the fibre input helps delay the onset of four-wave mixing effects and extend the measurement range.

The two optical tones are then amplitude-modulated by repeating pulses of 1 μs duration and 35 μs period, using a semiconductor optical amplifier that is driven by the output voltage of a pattern generator. The period of the pulses is longer than the two-way time of flight along the fibre under test. Pulses are then amplified by an erbium-doped fibre amplifier to an average output power of 5 mW with peak power on the order of 150 mW, and launched into a 3 km-long, standard single-mode fibre under test through a circulator. In few of the measurements sections of fibres with the coating removed, or fibres with polyimide coating, are connected as part of the fibre under test.

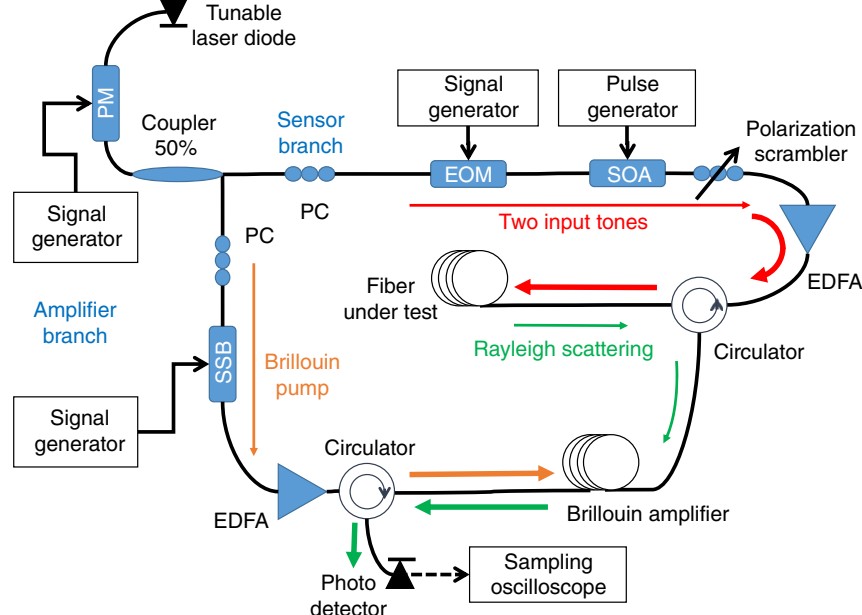

**Fig. 8** Schematic illustration of the experimental setup. Light from a tuneable laser diode source at a central optical frequency $\omega_0$ is used to generate all waveforms. An electro-optic phase modulator (PM) that is driven by a pseudo-random bit sequence is used to broaden the laser linewidth to the order of 100 MHz (broadening was only used in the experiments reported in Figs. 6, 7). Light in the sensor branch is modulated by an electro-optic amplitude modulator (EOM) and a semiconductor optical amplifier (SOA), to generate two tones at optical frequencies $\omega_0 \pm \frac{1}{2}\Omega$ within a common pulse envelope. The input waveform (red) is amplified by an erbium-doped fibre amplifier (EDFA) and launched into one end of a fibre under test. The two-field components are coupled by a forward stimulated Brillouin scattering process along the fibre, which strongly depends on the choice of the radio-frequency difference $\Omega$. Rayleigh back-scatter (green) of the two tones propagates back through a fibre-optic circulator into a second fibre section, which is used as a frequency-selective, backwards stimulated Brillouin scattering (SBS) amplifier. Light from the laser diode source in the amplifier branch (brown) is offset in frequency by a single-sideband electro-optic modulator (SSB), to generate the SBS pump wave. The SBS amplification is tuned to separate between the Rayleigh back-scatter contributions of the two optical tones. The amplified waveforms are detected by a photo-detector and sampled by a real-time oscilloscope for further off-line processing. Measurements are repeated for multiple choices of $\Omega$. The analysis of the collected Rayleigh back-scatter traces generates optomechanical time-domain reflectometry maps for distributed sensing of media outside the cladding (and even outside the coating) of the fibre under test. PC polarization controller

**Selective Brillouin amplification.** Rayleigh backscatter from the fibre under test propagates through the circulator into a second, 180 m-long fibre section, which serves as a narrowband backwards stimulated Brillouin scattering amplifier. Light in the second output path of the tuneable laser diode ('amplifier branch') is used as a backwards Brillouin pump wave. To that end, the laser diode output passes through a single-sideband, suppressed carrier electro-optic modulator, which is driven by a sine wave from a microwave generator. The microwave frequency $\Omega_{Pump}$ is adjusted to either $\Omega_B + \frac{1}{2}\Omega$ or $\Omega_B - \frac{1}{2}\Omega$, where $\Omega_B \approx 2\pi \times 10.85$ GHz is the Brillouin frequency shift in the amplifier section. The former setting provides selective Brillouin amplification of Rayleigh backscatter from the fibre under test at frequency $\omega_1 = \omega_0 + \frac{1}{2}\Omega$ only, whereas the latter provides similar amplification at frequency $\omega_2 = \omega_0 - \frac{1}{2}\Omega$ only. The Brillouin pump wave is amplified to 150 mW using a second erbium-doped fibre amplifier. Note that the amplifier branch passes through the input phase modulator. Hence any spectral broadening of the two input tones is also applied to the Brillouin pump wave.

The Brillouin amplifier operates in the linear regime, with optical power gain of only 5 dB. Due to the small gain, the optical waveform at the output of the amplifier also includes non-negligible contribution from the unamplified optical field component. In order to remove this background, reference traces are recorded with $\Omega_{Pump}$ detuned to $2\pi \times 12$ GHz. With this choice of frequency, the output waveform consists of unamplified contributions of both optical tones. These reference traces are subtracted from the amplified waveforms. The noise figure of Brillouin gain is on the order of 20 dB[60]. However, Brillouin amplification noise is mitigated by the averaging of repeating acquisitions (see also below). Compared with residual noise due to coherent Rayleigh backscatter in the fibre under test, the noise contribution of the Brillouin amplifier is not dominant. Rayleigh backscatter contributions from different positions along the fibre under test are characterized by different states of polarization, whereas Brillouin gain is highly polarization-dependent[61]. The state of polarization of the input pulses is therefore scrambled, in order to obtain uniform Brillouin amplification of backscatter contributions from all locations along the fibre under test.

**Detection and post processing.** The Brillouin-amplified, Rayleigh backscatter waveforms are detected by a photo-receiver of 200 MHz bandwidth, and sampled by a real-time digitizing oscilloscope at 5 ns intervals for further off-line processing. Each trace is first averaged over 64 repeating pulses. Measurements are then repeated over 1024 discrete choices of $\omega_0$, at wavelengths between 1558.5 nm and 1559.5 nm, in 1 pm increments. The wavelength range is extended to 2 nm in the measurements reported in Figs 6, 7 only. The Brillouin frequency shift $\Omega_B$ of the backwards Brillouin pump wave modulation is precisely adjusted for each $\omega_0$. Recorded traces are averaged over all choices of $\omega_0$ to reduce coherent Rayleigh backscatter noise. Lastly, the traces are spatially averaged by a moving window of 100 m width (50 m for Fig. 7).

We denote the processed traces of incoherent Rayleigh backscatter of the two waves from position $z$ as $\left\langle P_{1,2}^{(R)}(z) \right\rangle_\omega$. They are related to the corresponding incident power levels at the same location by: $\left\langle P_{1,2}^{(R)}(z) \right\rangle_\omega = C \exp(-\alpha z) \cdot P_{1,2}(z)$. The constant $C$ is governed by the recapture of Rayleigh backscatter and is independent of both $\omega_0$ and $\Omega$. The local nonlinear optomechanical coefficient is estimated offline according to Eq. (4):

$$\gamma^{(m)}(\Omega, z) = \frac{C \exp(-\alpha z)}{4 \left\langle P_1^{(R)}(z) \right\rangle_\omega \left\langle P_2^{(R)}(z) \right\rangle_\omega} \times$$
$$\left\{ \frac{d\left[ \left\langle P_2^{(R)}(z) \right\rangle_\omega - \left\langle P_1^{(R)}(z) \right\rangle_\omega \right]}{dz} + \alpha \left[ \left\langle P_2^{(R)}(z) \right\rangle_\omega - \left\langle P_1^{(R)}(z) \right\rangle_\omega \right] \right\}. \quad (5)$$

The measurement procedure is repeated for 40 choices of $\Omega$ within a range of $2\pi \times 40$ MHz, in order to construct the local optomechanical coupling spectra due to a radial acoustic mode $R_{0,m}$. The frequency step size is uneven. Small steps of $2\pi \times 0.25$ MHz are used near the resonance $\Omega_m$, whereas bigger steps up to $2\pi \times 2$ MHz are taken away from resonance towards the limits of the radio-frequency scanning range.

**Data availability.** The authors declare that the data supporting the findings of this study are available within the paper.

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

## Acknowledgements

The authors thank Dr. Yair Antman of Bar-Ilan University, (currently with Columbia University), for useful discussions. This work was supported in part by a Starter Grant from the European Research Council (ERC), grant number H2020-ERC-2015-STG 679228 (L-SID) and by the Israeli Science Foundation (ISF), grant number 1665-14.

## Author contributions

G.B., H.H.D., Y.L. and A.Z. performed mathematical analysis. G.B., H.H.D., Y.L. and E.P. designed the experimental setup, performed experiments and analysed data. A.Z. managed the project and wrote the paper.

## Additional information

**Competing interests:** The authors declare no competing interests.

