## [Peer Review File · Nature Communications]

Reviewers' comments:

Reviewer #1 (Remarks to the Author):

G. Bashan et al. present a new approach of distributed sensing which is based on forward Brillouin scattering (also known as guided acoustic wave Brillouin scattering - GAWBS). The novelty is to use *forward* scattering because established distributed sensing techniques (Raman, Rayleigh, Brillouin) are normally based on backward scattering which allows for time-of-flight measurements and are therefore easier to implement in a distributed fashion.

Using forward scattering for a distributed scan has not yet been shown before and is in general an interesting step forward for the fiber sensing community. The authors achieve a distributed measurement with 100m spatial resolution and show results of sensing different liquids outside the optical fiber and distinguishing different claddings. They also provide a theory model based on their own previous work.

Despite being an interesting work, I have several concerns and comments.

- One criticism to the authors is that they have already shown their concept of using GAWBS for sensing in terms of "listen where we cannot look" in a recent publication (Ref. [9]). Therefore, the novelty is the distributed scan, which achieves so far only 100m spatial resolution, which is not yet very useful. Moreover, it requires removing the cladding of the optical fiber, which makes the fiber very fragile (especially if 100m are required). Did the authors also try this measurement when removing the cladding over the 3km length of the fiber? What are the opportunities to improve the poor spatial resolution (which in terms of backscattering methods can be as high as on the meter-scale and further)?

- Another weakness of this method is that it is based on a difference in intensity and not in frequency. There are three points to be mentioned here. First of all, GAWBS are very unstable in intensity per se (a frequency shift can be measured much easier). Even if the authors say that also Rayleigh backscattering methods are based on intensity differences: forward scattering due to *transverse* acoustic modes is a lot noisier and weaker in its nature. Second, the technique itself measures the difference in intensity of the $-w$ and $+w$ Rayleigh backscattered signal (+noise), which itself was amplified by backward Brillouin scattering (+noise). This results in uncertainties in the intensity. Moreover, the $-w$ and $+w$ component are measured one after the other, which also inserts more noise. Third, the difference of the different liquids is made by comparing the intensity of the backscattered signal. I have therefore some doubts if this has a lot of potential in terms of distributed sensing.

- The authors provide some theory in the paper and the supplementary. Here, they mostly cite their previous work. That is fair enough but raises two things: it would be good to base the theory part also on other publications (there is quite some theory out there in different sensor groups). Second, if they only cite themselves, the question can be asked, if this is interesting also for the broader community?

- It would be good to have a Figure 0, which explains the general concept. It is not clear from the beginning how the spatial distribution is achieved. They use a combination of Rayleigh scattering, GAWBS and then amplify the obtained signal selectively with backward Brillouin scattering. It is not an easy concept and the paper would benefit from a first overview figure.

- It is mentioned "1024 choices of w_0 ". Does that mean discrete values over the wavelength span?

- I'm surprised that the average output power of the pulses is only 5mW (using 1us pulses). These are SMF fibers and 3km long. Compared to the forward scattering experiments in Russells group

[31,etc] where high-power ps-pulses were used to pump GAWBS, this seems to be very low in order to excite transverse acoustic modes, especially as the acousto-optic overlap is less high than in photonic crystal fibers.

- Some minor things: it would be good to mention the frequency resolution (instead of choices of 40 or 1024 wavelengths). Also, most of the papers cited are older ones and the ones from Russell's group or themselves. There are also other active groups in this field, as in Brazil ("Brillouin scattering self-cancellation"), France and other places.

In conclusion, the paper represents a very interesting work and is the first to show distributed sensing based on GAWBS. However, the spatial resolution of 100m is not yet an excellent result and the fact that the sensor is based on intensity differences is a weak point and makes it difficult to judge the potential. Given the broad scope of Nature Communications, this work would maybe be better suited for a more specialized journal. I think that the novelty is there but the impact might be limited.

Reviewer #2 (Remarks to the Author):

Comment on the paper : « " Opto-Mechanical Time-Domain Reflectometry," by Gil Bashan et al. The paper reports an interesting work on distributed measurements using forward Brillouin scattering from radial acoustic waves in optical fibers. It builds on the previous work of the same authors on the measurement of the mechanical impedance of liquids outside the cladding of an optical fiber (See Ref. 9). In the previous work, the authors demonstrated liquid sensing outside optical fibers using forward stimulated Brillouin scattering. In the present work, the authors demonstrate a distributed measurement of this effect using Rayleigh backscattering and Brillouin amplification. This is achieved over 3 km of standard single-mode fibers with 100 m resolution. The paper is well written and the results demonstrated in this work may add a new dimension for distributed measurements outside the optical fibers, and may intrigue new possibilities for technological applications of optical fibers. I recommend its publication in Nature Communications, after addressing the following concerns:

1- Figure 4 is not really convincing for potential applications to the detection of liquids.

Could the author providing more details about the measurement results ? e.g., the sensitivity, the accuracy, and the efficiency of the method compared to standard distributed measurement techniques.

2- Why the spatial resolution is so bad ? why not increase it by reducing the pulse duration ?

3- How much is the gain and noise of the Brillouin amplifier ?

4- In the introduction, the authors write : "The principle is based on guided acoustic wave Brillouin scattering: a nonlinear propagation effect that couples between guided light waves and guided acoustic modes of the fibre". As initially stated by Shelby et al. in 1985, GAWBS is a pure linear spontaneous scattering effect arising from thermal acoustic phonons with a scattering efficiency that linearly depends on the pump power. It can only become nonlinear when it is stimulated. In other words, stimulated forward Brillouin scattering is a nonlinear effect while GAWBS is not.

Reviewer #3 (Remarks to the Author):

Opto-mechanical interactions are currently a flourishing topic of research and have given rise to many applications, thanks to the remarkably strong coupling between optical fields and material vibrations. The authors have recently proposed [Ref 9] to exploit such interactions in a very innovative configuration to sense properties of the material outside the fibre (in the present case the acoustic impedance of this material) while the light keeps confined in the fibre core, so with no contact between the guided light and the outer material. This is a very smart approach and certainly a breakthrough in chemical sensing based on optical fibres.

In this manuscript the authors present a significant progress by making the sensing system distributed. To my knowledge this is the first demonstration of a distributed implementation of this novel sensing technique, though it is known that at least 2 other competing teams are active to reach this objective, one of them even presenting partial results in a conference. This first real demonstration has to be credited to the authors, tempered by the fact that only 30 resolved points are obtained and the usefulness of such a modest performance has to be seriously questioned. The main issue is that no significant margin of progression is foreseen from the proposed technique, for quite fundamental reasons, and no suggestion for improvement is made by the authors. This puts a shade on the real impact of the technique.

The paper is perfectly well composed in a clear and fluid language, with good quality graphical material. The essential information is presented in a condensed style in the main body of the article, while a high quality and solid justification of the physical principles, together with a rigorous derivation of the main equations is presented in the supplementary material.

To clearly position the paper, I would appreciate if the authors can address the points I could raise after a thorough examination of the manuscript:

- 1) (Minor) Figure 1: the dashed lines do not appear as clearly "dashed" and can be hardly distinguished from the solid lines. Is it possible to clarify why the 2 distributions with no interaction (dashed lines) are slightly but clearly dissimilar, like experiencing a different attenuation?
- 2) (Minor) The authors extensively use the trendy term "novel paradigm" to designate their realisation. The terms used in a Nature publication must be accurate and paradigm sounds very inappropriate to designate a novel implementation, moreover using techniques that were already validated in another context (see next comment). Paradigm is clearly defined in philosophy and epistemology and, as far as it can be seen, the proposed system does not imply a novel conception of the notion of sensing. I suggest to the authors to change any occurrence of "paradigm" into "configuration" or "implementation", this won't decrease the impact of their statements.
- 3) (Major) Rayleigh reflectometry for mapping nonlinear interactions has been used for some 20 years, particularly for 4-wave mixing to deduce local values of chromatic dispersion (doi:10.1364/OL.21.001724). This work must preferably be cited, since it is prior art. The implementation realized by the authors was substantially more difficult regarding the very small spectral separation between the interacting signals. Nevertheless the principle is very similar and both implementations are subject to the same limitations.
- 4) (Major) The authors must discuss much more in details the limitations. They mention honestly that 4-wave mixing is setting a limitation for the range, but without developing the issue. Actually this is a serious limitation and it gives little hope to much improve the technique. In particular, the spatial resolution must be absolutely improved over a similar distance range, which is normally obtained by compensating the shorter pulse by a larger power. However, this will shorten the range as a result of the detrimental impact of 4-wave mixing and possibly modulation instability too, so that increasing the number of resolved points looks vain at first glance. This issue must be certainly addressed in more details to clarify the impact of the technique. The authors must also address realistic possibilities to improve the technique, if ever there are any. They keep surprisingly silent on this point.
- 5) (Minor) Line 91: "suppresses" is inaccurate, since the statistical nature of coherent Rayleigh scattering makes it suppressed only with an infinite ensemble. The requirement of averaging the signal over plenty of wavelengths is a strong penalty for the technique and weakens its impact.
- 6) (Medium) Line 137-138: The statement sounds inaccurate, since in traditional OTDRs there is no derivative taken over the data. The authors must clarify the analogy made in their statement.
- 7) (Minor) Line 166: why mentioning the average power and not the peak power? The peak power

will certainly matter much more for the limitations and the response and is therefore more informative for the reader.

8) (Medium) Line 179-180: The gain provided by the Brillouin amplifier is certainly large by a simple estimation (not mentioned, but would be informative), so is it really useful to insert a polarisation scrambler? Will the polarisation of the signal be pulled to the pump polarisation [ref 43], so that the input signal polarisation has little impact on the final amplification?

In response to the comments of Reviewer 1:

*"G. Bashan et al. present a new approach of distributed sensing which is based on forward Brillouin scattering (also known as guided acoustic wave Brillouin scattering - GAWBS). The novelty is to use *forward* scattering because established distributed sensing techniques (Raman, Rayleigh, Brillouin) are normally based on backward scattering which allows for time-of-flight measurements and are therefore easier to implement in a distributed fashion. Using forward scattering for a distributed scan has not yet been shown before and is in general an interesting step forward for the fiber sensing community. The authors achieve a distributed measurement with 100m spatial resolution and show results of sensing different liquids outside the optical fiber and distinguishing different claddings. They also provide a theory model based on their own previous work."*

Reply: We thank the reviewer for finding our work to be an interesting step forward. We do wish to point out, however, that the novelty of this work is not just in the distributed monitoring of a forward-scattering process. **The main novelty is in the distributed analysis of media outside the cladding of fibers, and even outside coated fibers** (see also below). The concluding paragraph of the manuscript was revised to stress out this point.

Comment: *"...weakness of this method is that it is based on a difference in intensity and not in frequency. There are three points to be mentioned here. First of all, GAWBS are very unstable in intensity per se (a frequency shift can be measured much easier). Even if the authors say that also Rayleigh backscattering methods are based on intensity differences: forward scattering due to *transverse* acoustic modes is a lot noisier and weaker in its nature. Second, the technique itself measures the difference in intensity of the $-w$ and $+w$ Rayleigh backscattered signal (+noise), which itself was amplified by backward Brillouin scattering (+noise). This results in uncertainties in the intensity. Moreover, the $-w$ and $+w$ component are measured one after the other, which also inserts more noise. Third, the difference of the different liquids is made by comparing the intensity of the backscattered signal. I have therefore some doubts if this has a lot of potential in terms of distributed sensing."*

Reply: We respect the reviewer's perspective. However, as detailed below, we must say that **this comment represents a fundamental misunderstanding of the proposed concept. We distinguish between media based on the LINEWIDTH of stimulated acoustic waves, and NOT BASED ON INTENSITY.** While the responsibility for this misunderstanding might also be ours, at least in part, it must nevertheless be corrected. **The distinction between measurements of intensity and linewidth is essential to the evaluation of our manuscript.** Its clarification would take away much of the grounds for the reviewer's criticism.

The acoustic waves' spectra are indeed constructed using multiple measurements of differential intensity. However, **the decision variable is the spectral linewidth. The linewidth measurement is far more robust to noise than intensity readings. Figures 3 and 4, (now Figures 4 and 5), clearly show measurements of gain spectra and linewidths.** This comment therefore comes as somewhat of a surprise to us. We added a new Figure 1 to the manuscript (see also below), to introduce the concept

and explain the measurement of spectral linewidth. The distinction between intensity and linewidth measurement is also highlighted in the text.

We do not understand why the reviewer claims that GAWBS is "unstable in intensity per se". This has not been our experience. We also disagree with the claim that forward scatter due to GAWBS is "noisier and weaker in nature" than Rayleigh back-scatter. Our measurements suggest the opposite: the change in power per unit length due to GAWBS can actually be larger than that induced by Rayleigh scattering.

The reviewer doubts the potential for future improvements in the distributed sensing performance. **However, these doubts are affected by misinterpretation of our work.** We agree that the present performance must be improved to meet the requirements of most applications. That being said, several potential directions for performance enhancement may already be identified (see below).

Comment: "...the distributed scan, which achieves so far only 100m spatial resolution, which is not yet very useful. What are the opportunities to improve the poor spatial resolution (which in terms of backscattering methods can be as high as on the meter-scale and further)?"

Reply: We agree with the reviewer that the current performance of 100 m resolution is modest. **However, this limitation is not fundamental. Several potential avenues for performance enhancement may be identified.** Double-pulse measurement techniques, which are successfully employed in Brillouin sensors, can improve OM-TDR resolution beyond the limitations of acoustic lifetimes. The range of optical frequencies that are used in the averaging of coherent Rayleigh back-scatter may be extended, to provide better signal-to-noise ratios. The protocol can be extended to the monitoring of multiple scattering sidebands, supporting a longer range and higher pump power. The onset of scattering to higher orders may be delayed using unequal power levels of the two input field components. The use of coated fibers provides additional opportunities: the coating impedance and diameter may be optimized for improved sensitivity of detection of specific target media. These prospects are discussed in the revised manuscript.

The reviewer draws a comparison between our methods and present-day, state-of-the-art distributed sensors that rely on back-scattering. However, such comparison is irrelevant. As an alternative comparison, we may note that the early papers on distributed Brillouin sensing from 1989 and 1990 presented measurements over 1.2 km of fiber with 100 meters resolution. Those papers nevertheless started an entire field of research and applications. Resolution has since improved by a factor of 100,000 and range increased over 100-fold. We doubt whether such improvements could have been foreseen at the time of those first papers. Of course, at this stage we do not compare the value of our methods with the large, proven impact of Brillouin sensors. Not even remotely. However, **the potential impact of our new technique should not be ruled out based on its current performance alone.**

Comment: (The method) "...requires removing the cladding of the optical fiber, which makes the fiber very fragile (especially if 100m are required). Did the authors also try this measurement when removing the cladding over the 3km length of the fiber?"

Reply: **This comment is incorrect. The manuscript also reports measurements taken outside commercially-available, polyimide-coated fibers (Figure 5, which is now Figure 6).** This result is mentioned several times in the text. We are surprised that the reviewer failed to notice it. **The solution we propose is sensing outside coated fibers.** Bare fibers with no coating only serve as means for laboratory research. The abstract and body text were modified to emphasize this point even more clearly.

Comment: "- One critics to the authors is that they have already shown their concept of using GAWBS for sensing in terms of 'listen where we cannot look' in a recent publication (Ref. [9])."

Reply: The current manuscript is indeed the next step following ref. [9], in which *point-sensing* based on GAWBS was demonstrated. However, the principle used in the current manuscript is different: In [9], guided acoustic waves were stimulated by the spectral components of pump light at one wavelength, whereas a probe wave of a different wavelength was scattered by the same acoustic waves. The probe did not affect the acoustic waves. **Hence the scattering of the probe wave in [9] could be regarded as a spontaneous process. Here, two optical tones stimulate the acoustic waves, and we monitor the counter-effect on the same two tones. The process is therefore stimulated** (see also a comment by Reviewer 2). The revised manuscript was modified to clarify this distinction. In addition, ref. [9] only addressed bare fibers whereas the current manuscript also reports measurements outside coated fibers (see also above).

Comment: "- The authors provide some theory in the paper and the supplementary. Here, they mostly cite their previous work. That is fair enough but raises two things: it would be good to base the theory part also on other publications (there is quite some theory out there in different sensor groups). Second, if they only cite themselves, the question can be asked, if this is interesting also for the broader community? ... Also, most of the papers cited are older ones and the ones from Russell's group or themselves. There are also other active groups in this field, as in Brazil ("Brillouin scattering self-cancellation"), France and other places."

Reply: We have the highest regards for many colleagues doing excellent research on GAWBS and related phenomena. We apologize if the original list of references has not been sufficiently comprehensive. Several references were added to the main text and supplementary information.

The initial proposition of GAWBS-based sensing [9] raised large interest in the community. This interest manifests in citations and conference talks. As noted by Reviewer 3, the interest of the community is also made apparent by several competing efforts. We refer to these efforts in the revised manuscript.

Comment: "It would be good to have a Figure 0, which explains the general concept. It is not clear from the beginning how the spatial distribution is achieved. They use a combination of Rayleigh scattering, GAWBS and then amplify the obtained signal selectively with backward Brillouin scattering. It is not an easy concept and the paper would benefit from a first overview figure."

Reply: We thank the reviewer for this comment. An introductory Figure 1 was added to the revised manuscript. The principle of operation is also explained in greater detail in the text.

Comment: "- It is mentioned "1024 choices of w_0 ". Does that mean discrete values over the wavelength span?"

Reply: The reviewer is correct. 1,024 discrete values of the central optical frequency were used. This point is clarified in the revised manuscript.

Comment: "- I'm surprised that the average output power of the pulses is only 5mW (using 1us pulses). These are SMF fibers and 3km long. Compared to the forward scattering experiments in Russells group [31,etc] where high-power ps-pulses were used to pump GAWBS, this seems to be very low in order to excite transverse acoustic modes, especially as the acousto-optic overlap is less high than in photonic crystal fibers."

Reply: The peak power level of the pump pulses was on the order of 150 mW. The peak power is noted in the revised manuscript. This peak power level is in agreement with previous characterization of GAWBS in standard single-mode fibers (see for example [40]). Note that the GAWBS interaction accumulates over resolution segments that are 100 meters-long.

Comment: "- Some minor things: it would be good to mention the frequency resolution (instead of choices of 40 or 1024 wavelengths)."

Reply: The scanning of the central optical frequency was carried out in wavelength increments of 1 pm. This step-size is noted in the revised manuscript. The radio-frequency step-size was not uniform. Smaller steps of 0.25 MHz were used near the GAWBS resonance, whereas larger steps of 2 MHz were taken further away from the resonance condition. The scan is described in the Methods Section of the revised manuscript.

Comment: "In conclusion, the paper represents a very interesting work and is the first to show distributed sensing based on GAWBS. However, the spatial resolution of 100m is not yet an excellent result and the fact that the sensor is based on intensity differences is a weak point and makes it difficult to judge the potential. Given the broad scope of Nature Communications, this work would maybe be better suited for a more specialized journal. I think that the novelty is there but the impact might be limited."

Reply: We thank the reviewer for expressing interest in our work, and we are glad that he/she acknowledges the novelty of distributed GAWBS sensing. We fully agree with the reviewer and accept that the current spatial resolution and number of resolution points are modest. However, **we object his/her bottom-line conclusion.** As noted above, the sensor distinguishes among media based on the reconstruction of spectral linewidths and NOT based on intensity, a principle that was overlooked by the reviewer. **The current performance cannot be considered as a restrictive bound: several potential directions for future performance improvements have been identified, as discussed above. The measurements outside coated fibers, which were also overlooked, further add to the**

potential impact of the proposed principle. Interest in this work is not restricted to fiber-sensor specialists only: it is also highly relevant in the broader contexts of opto-mechanics, nonlinear optics and fiber-optics.

We believe that the clarifications provided in this reply letter and in the revised manuscript could affect the conclusion of the reviewer.

In response to the comments of Reviewer 2:

"The paper reports an interesting work on distributed measurements using forward Brillouin scattering from radial acoustic waves in optical fibers. It builds on the previous work of the same authors on the measurement of the mechanical impedance of liquids outside the cladding of an optical fiber (See Ref. 9). In the previous work, the authors demonstrated liquid sensing outside optical fibers using forward stimulated Brillouin scattering. In the present work, the authors demonstrate a distributed measurement of this effect using Rayleigh backscattering and Brillouin amplification. This is achieved over 3 km of standard single-mode fibers with 100 m resolution. The paper is well written and the results demonstrated in this work may add a new dimension for distributed measurements outside the optical fibers, and may intrigue new possibilities for technological applications of optical fibers. I recommend its publication in Nature Communications, after addressing the following concerns:"

Reply: We thank the reviewer for his/her support of our work.

Comment 1: "Figure 4 is not really convincing for potential applications to the detection of liquids. Could the author providing more details about the measurement results? e.g., the sensitivity, the accuracy, and the efficiency of the method compared to standard distributed measurement techniques."

Reply: Figure 4 (now Figure 5) was replaced in the revised manuscript. The comparison between the performance of our methods and those of other techniques is difficult to draw, as no equivalent techniques were reported to-date. Point-sensors based on optical fibers can measure the refractive index of an outside medium under test with $1e-7$ RIU precision. However, no such method can be scaled towards distributed analysis. The precision of our measurements is expressed in terms of the uncertainty in the estimated opto-mechanical linewidth. In the revised manuscript, experimental uncertainty is also quantified in terms of mechanical impedance.

Comment 2: "Why the spatial resolution is so bad? Why not increase it by reducing the pulse duration?"

Reply: The spatial resolution is indeed modest. Resolution enhancement through the use of shorter pulses is restricted by two mechanisms: the lifetime associated with the acoustic oscillations, and the signal-to-noise ratio (SNR) of measurements. Of the two limitations, that of the SNR is currently the more severe. Prospects for resolution enhancement are discussed in the revised manuscript. These include the incorporation of double-pulse techniques that were developed for Brillouin analysis, broader scanning of the optical frequencies, and higher pump power levels in conjunction with monitoring of multiple scattering orders. We strongly believe that the current spatial resolution can and will be improved in future studies.

Comment 3: "How much is the gain and noise of the Brillouin amplifier?"

Reply: The Brillouin amplifier was adjusted for a modest gain of 5 dB. It is operating in the linear regime, and its noise figure is on the order of 100. Noise due to the Brillouin amplifier was effectively averaged out over multiple acquisitions. Compared with residual noise due to coherent Rayleigh backscatter, the noise contribution of the Brillouin amplifier is not dominant. The above details of Brillouin amplification were added to the Methods Section of the revised manuscript.

Comment 4: "In the introduction, the authors write: "The principle is based on guided acoustic wave Brillouin scattering: a nonlinear propagation effect that couples between guided light waves and guided acoustic modes of the fibre". As initially stated by Shelby et al. in 1985, GAWBS is a pure linear spontaneous scattering effect arising from thermal acoustic phonons with a scattering efficiency that linearly depends on the pump power. It can only become nonlinear when it is stimulated. In other words, stimulated forward Brillouin scattering is a nonlinear effect while GAWBS is not."

Reply: We thank the reviewer for this comment. Better distinction between the discussion of stimulated and spontaneous processes involving guided acoustic waves is made in the revised manuscript and Supplementary Material.

In response to the comments of Reviewer 3:

"Opto-mechanical interactions are currently a flourishing topic of research and have given rise to many applications, thanks to the remarkably strong coupling between optical fields and material vibrations. The authors have recently proposed [Ref 9] to exploit such interactions in a very innovative configuration to sense properties of the material outside the fibre (in the present case the acoustic impedance of this material) while the light keeps confined in the fibre core, so with no contact between the guided light and the outer material. This is a very smart approach and certainly a breakthrough in chemical sensing based on optical fibres."

"In this manuscript the authors present a significant progress by making the sensing system distributed. To my knowledge this is the first demonstration of a distributed implementation of this novel sensing technique, though it is known that at least 2 other competing teams are active to reach this objective, one of them even presenting partial results in a conference. This first real demonstration has to be credited to the authors, tempered by the fact that only 30 resolved points are obtained and the usefulness of such a modest performance has to be seriously questioned. The main issue is that no significant margin of progression is foreseen from the proposed technique, for quite fundamental reasons, and no suggestion for improvement is made by the authors. This puts a shade on the real impact of the technique."

"The paper is perfectly well composed in a clear and fluid language, with good quality graphical material. The essential information is presented in a condensed style in the main body of the article, while a high quality and solid justification of the physical principles, together with a rigorous derivation of the main equations is presented in the supplementary material."

Reply: We thank the reviewer for his/her evaluation of our former and current work. We are glad that the reviewer considers the current manuscript as a significant and novel progress. We also agree with the reviewer that the current number of resolution points is modest and insufficient for most practical purposes. However, **as discussed in detail below, it is our opinion that progress can be made, and that the impact of the technique cannot be evaluated based on its present performance alone.**

We address the two major comments of the reviewer first.

Comment 3) (Major): "Rayleigh reflectometry for mapping nonlinear interactions has been used for some 20 years, particularly for 4-wave mixing to deduce local values of chromatic dispersion (doi:10.1364/OL.21.001724). This work must preferably be cited, since it is prior art. The implementation realized by the authors was substantially more difficult regarding the very small spectral separation between the interacting signals. Nevertheless the principle is very similar and both implementations are subject to the same limitations."

Reply: We thank the reviewer for this comment. We were unaware of this significant prior art. This important reference is acknowledged in the revised manuscript [51]. A significant follow-up, in which Brillouin analysis was used to monitor four-wave-mixing components, is acknowledged as well [52].

Comment 4) (Major): "The authors must discuss much more in details the limitations. They mention honestly that 4-wave mixing is setting a limitation for the range, but without developing the issue. Actually this is a serious limitation and it gives little hope to much improve the technique. In particular, the spatial resolution must be absolutely improved over a similar distance range, which is normally obtained by compensating the shorter pulse by a larger power. However, this will shorten the range as a result of the detrimental impact of 4-wave mixing and possibly modulation instability too, so that increasing the number of resolved points looks vain at first glance. This issue must be certainly addressed in more details to clarify the impact of the technique. The authors must also address realistic possibilities to improve the technique, if ever there are any. They keep surprisingly silent on this point."

Reply: We fully agree with the reviewer that the currently-achieved spatial resolution is modest, and insufficient for most applications. The discussion of resolution and range limitations was extended in the revised manuscript, as the reviewer requires. However, **we do not share the reviewer's conviction that resolution has little room for improvement.** The onset of four-wave mixing may be delayed further if the input power levels of the two optical tones are made unequal: a stronger "pump" (higher-frequency tone) and a weaker "signal" (lower-frequency tone), in analogy to Brillouin sensors. The measurement protocol can be extended to monitor and account for multiple scattering orders. The measurement signal-to-noise ratio can be improved with the scanning of the central optical frequency over a wider range. Double-pulse schemes, borrowed again from the realm of Brillouin sensors, may be used to alleviate the resolution limitations that are associated with acoustic lifetimes. Lastly, the

coating diameter and composition can be optimized for sensitivity enhancements. These prospects are addressed in a discussion that is added to the revised manuscript.

We address next the other comments that were included in the review.

Comment 1) (Minor): "Figure 1. The dashed lines do not appear as clearly "dashed" and can be hardly distinguished from the solid lines. Is it possible to clarify why the 2 distributions with no interaction (dashed lines) are slightly but clearly dissimilar, like experiencing a different attenuation?"

Reply: Figure 1 (which is currently Figure 2) was replaced by a clearer version. The reviewer is right to note that the attenuation levels of the two dashed line traces are indeed slightly different. The pair of traces was taken with a frequency offset of 340 MHz, which happens to be near the resonance of a torsional-radial guided acoustic mode of the fiber (*TR* mode). While the stimulation of *TR* modes is much less efficient than that of the radial modes, it is nevertheless nonzero. This residual coupling off the resonance of the radial mode does not affect the linewidth measurements, as can be seen in Figures 3 through 6. This discussion is added to the revised manuscript.

Comment 2) (Minor): "The authors extensively use the trendy term 'novel paradigm' to designate their realisation. The terms used in a Nature publication must be accurate and paradigm sounds very inappropriate to designate a novel implementation, moreover using techniques that were already validated in another context (see next comment). Paradigm is clearly defined in philosophy and epistemology and, as far as it can be seen, the proposed system does not imply a novel conception of the notion of sensing. I suggest to the authors to change any occurrence of 'paradigm' into 'configuration' or 'implementation', this won't decrease the impact of their statements."

Reply: The term "paradigm" was removed from the revised manuscript.

Comment 5) (Minor): "Line 91: 'Suppresses' is inaccurate, since the statistical nature of coherent Rayleigh scattering makes it suppressed only with an infinite ensemble. The requirement of averaging the signal over plenty of wavelengths is a strong penalty for the technique and weakens its impact."

Reply: We agree with the reviewer that coherent Rayleigh scattering noise can be mitigated, but not suppressed entirely. This is clarified in the revised manuscript. The repeating measurements over many optical frequencies are indeed a weakness. Scanning a broader range of optical frequencies may help reduce the number of traces that is necessary. This note was also added to the revised manuscript.

Comment 6) (Medium): "Line 137-138. The statement sounds inaccurate, since in traditional OTDRs there is no derivative taken over the data. The authors must clarify the analogy made in their statement."

Reply: The analogy relates to the fact that estimates of local losses based on traditional OTDRs require taking the derivative of data. However, this argument is not central to the discussion. It was therefore removed from the revised manuscript, to avoid unnecessary controversy.

Comment 7) (Minor): "Line 166. Why mentioning the average power and not the peak power? The peak power will certainly matter much more for the limitations and the response and is therefore more informative for the reader."

Reply: The peak power of pulses was on the order of 150 mW. It is noted in the revised manuscript.

Comment 8) (Medium): "Line 179-180. The gain provided by the Brillouin amplifier is certainly large by a simple estimation (not mentioned, but would be informative), so is it really useful to insert a polarisation scrambler? Will the polarisation of the signal be pulled to the pump polarisation [ref 43], so that the input signal polarisation has little impact on the final amplification?"

Reply: The power gain of the Brillouin amplifier was actually quite small: only 5 dB. This is noted in the Methods Section of the revised manuscript. Polarization pulling at such gain is rather weak. Rayleigh backscatter contributions from different positions along the fiber under test have different states of polarization. The scrambling of signal polarization is necessary in order to obtain equal Brillouin amplification of Rayleigh backscatter from all locations along the fiber under test. This issue is clarified in the revised manuscript as well.

Reviewers' comments:

Reviewer #1 (Remarks to the Author):

The authors replied to the reviewers' comments and revised their manuscript accordingly. I appreciate the new figure 1, I think it helps the reader to understand the concept.

I agree that the measurement of the linewidth is more robust than the intensity. However, my critics relied more on the fact that the measurement method relies on the difference of intensity of two Rayleigh backscattered signals, which are amplified by Brillouin scattering, plus the fact that the measurements are taken one after the other and not at the same time. I do not doubt that it is an interesting measurement as I already pointed out. However, I still believe that the measurement in this fashion has severe limitations (in accuracy and spatial resolution as also pointed out by reviewer 3).

As the sensing of liquid outside of an optical fiber has already been shown in Ref. [9], I had considered the novelty to be the distributed measurement of forward Brillouin scattering.

Concerning the spatial resolution, it is true that distributed sensors based on backward scattering also started with a modest value, however it is not 'irrelevant' to compare a potentially new distributed sensing system to nowadays state-of-the-art. Therefore, pointing out the limitations and opportunities is crucial (the authors added a paragraph on that, which is good).

Concerning the comment on the coating of the fibre (now figure 6), just to understand it correctly: why was a 200m-piece with polyimide coating spliced in between normal fibers instead of using a uniform fiber all along or using a standard fiber as a sensing element? This had triggered my previous comment on removing or replacing the coating of a standard fiber.

As I already commented before: the novelty is there, measuring forward Brillouin scattering in a distributed fashion is definitely interesting, though the spatial resolution is very modest and the impact of this measurement method for real applications might be limited.

Reviewer #2 (Remarks to the Author):

Considering that the decision of the Editor to accept a revision, I find that the authors addressed correctly the concerns and the questions of the reviewers in this revised version and in the "Response to reviewers". The authors greatly improved their manuscript and the drawn conclusions are now well supported by their data and simulations. I believe that despite the mentioned weakness about spatial resolution of this new method, this paper can be published in Nature Communications as it may represent a new class of distributed optical fiber sensors with strong potential applications.

Reviewer #3 (Remarks to the Author):

I would like to personally thank the authors for their efforts to respond to the reviewers' questions. Their replies are clear and they could in the vast majority of cases argue with convincing statements. I appreciate that they emphasize on the difficulty to realize distributed measurements using a forward scattering process: this was only realized in a few cases in the past (now properly cited) and it requires inventive qualities to combine several fairly distinct processes, resulting in a very complex measurement system. This must definitely be credited to the authors.

They did actually respond satisfactorily to all my questions, except that they could not fully convince me about the reality of a large margin for improvement.

The idea to use a double-pulse technique to improve the spatial resolution is at first glance conceptually correct and makes fully sense. However, the response will be reduced by a proportional amount and here I don't see how they can realistically compensate this attenuated response. If we assume a 10-times better spatial resolution - which is a reasonable target to make the system practical - the noise and fluctuations must be reduced in the same proportion, or the response increased in the same proportion, too.

Practically it means to take 100X more wavelength averaging to reduce the coherent Rayleigh noise, which sounds not very realistic. Another option would be to increase the power in the signal by a factor 10 (assuming an additive noise), but the waves will be rapidly depleted by 4-wave mixing, so this is not realistic as well.

I have to say that the proposition to make the 2 optical tones unequal in power to reduce 4-wave mixing is totally incorrect: if P_1 is reduced by a factor N and P_2 is multiplied by the same factor N , the interacting waves are of unequal power while the response keeps identical since given by the product $P_1 \times P_2$ according to Equ. S20-S21. But it is straightforward to show that the total power summed in the two 4-wave mixing sidebands is minimum when $N=1$, so when the powers are equal.

That's why I keep convinced that the proposed configuration has little margin for improvement, unless an entirely different scheme is used to retrieve the distributed power of the interacting waves.

I totally follow the authors when they say it is a first step and they make the comparison with the development of distributed Brillouin sensing which reached similarly poor distributed performance in the first publication. We can also counter-argue that we are 30 years later and there was a tremendous improvement in the response of the distributed sensing techniques in the meantime, so today's tools are much better sharpened.

As I said in my introduction I am fully satisfied by the other responses given by the authors. There are just 2 points on which I need further clarification:

1) In Figure 2, why the amplification of the lower frequency is smaller than the attenuation of the higher frequency, as a result of the interaction? They must exactly compensate according to the equations derived by the authors, but there is nearly a factor 2 which cannot be reasonably credited to experimental artefacts.

2) In their response the authors clarify that the gain of their frequency-selective Brillouin amplifier is 5 dB, which is just a factor 3. So it means that there is a non-negligible contribution from the other tone in the recorded trace. This must be mentioned and the impact evaluated.

In response to the comments of Reviewer 1:

"The authors replied to the reviewers' comments and revised their manuscript accordingly. I appreciate the new figure 1, I think it helps the reader to understand the concept."

Reply: We thank the reviewer again for his suggestion to add Fig. 1.

Comment: *"I agree that the measurement of the linewidth is more robust than the intensity. However, my critics relied more on the fact that the measurement method relies on the difference of intensity of two Rayleigh backscattered signals, which are amplified by Brillouin scattering, plus the fact that the measurements are taken one after the other and not at the same time. I do not doubt that it is an interesting measurement as I already pointed out. However, I still believe that the measurement in this fashion has severe limitations (in accuracy and spatial resolution as also pointed out by reviewer 3)..."*

"...As I already commented before: the novelty is there, measuring forward Brillouin scattering in a distributed fashion is definitively interesting, though the spatial resolution is very modest and the impact of this measurement method for real applications might be limited."

Reply: We are pleased that the reviewer finds that our work is novel and interesting. We are also glad that the reviewer acknowledges the significance of linewidth measurements.

We completely agree with the reviewer that the main challenge to our method is the signal-to-noise ratio (SNR) of processed traces, and that the current spatial resolution is modest. However, **we maintain that the performance of our sensing concept can and will be improved with more time and future efforts.** Such progress extends beyond a single work. In support of this point of view, **the revised manuscript describes a breakthrough in the measurement SNR**, which is achieved without increase in the number of averages. Improvement is based on the spectral broadening of the laser source, from sub-MHz linewidth to 100 MHz, using external phase modulation. Use of a broadened source reduces noise due to coherent Rayleigh backscatter in each raw data trace.

The higher SNR is leveraged towards **measurements with 50 m resolution, twice higher than we could previously obtain.** The publication of our results would lead more groups to follow similar methods, and push performance even further. The principles leading to higher SNR and measurements with higher resolution are reported in the revised manuscript, in the Main Text and in the Methods section.

Comment: *"As the sensing of liquid outside of an optical fiber has already been shown in Ref. [9], I had considered the novelty to be the distributed measurement of forward Brillouin scattering. Concerning the spatial resolution, it is true that distributed sensors based on backward scattering also started with a modest value, however it is not 'irrelevant' to compare a potentially new distributed sensing system to nowadays state-of-the-art. Therefore, pointing out the limitations and opportunities is crucial (the authors added a paragraph on that, which is good)."*

Reply: Again, we are glad that reviewer finds the discussion of limitations and opportunities to be effective. Regarding spatial resolution, please see above our reply to the previous comment.

Comment: *"Concerning the comment on the coating of the fibre (now figure 6), just to understand it correctly: why was a 200m-piece with polyimide coating spliced in between normal fibers instead of*

using a uniform fiber all along or using a standard fiber as a sensing element? This had triggered my previous comment on removing or replacing the coating of a standard fiber."

Reply: We agree with the reviewer that long, uniform sections of polyimide-coated fiber are a promising platform for our proposed protocol. We were recently able to obtain such sections, which were unavailable to us at the time of the previous submission. **Fig. 6 of the revised manuscript was replaced by preliminary distributed analysis over 2.5 km of polyimide-coated fiber.**

In response to the comments of Reviewer 2:

"Considering that the decision of the Editor to accept a revision, I find that the authors addressed correctly the concerns and the questions of the reviewers in this revised version and in the "Response to reviewers". The authors greatly improved their manuscript and the drawn conclusions are now well supported by their data and simulations. I believe that despite the mentioned weakness about spatial resolution of this new method, this paper can be published in Nature Communications as it may represent a new class of distributed optical fiber sensors with strong potential applications."

Reply: We thank the reviewer for his/her support of our work.

In response to the comments of Reviewer 3:

"I would like to personally thank the authors for their efforts to respond to the reviewers' questions. Their replies are clear and they could in the vast majority of cases argue with convincing statements. I appreciate that they emphasize on the difficulty to realize distributed measurements using a forward scattering process: this was only realized in a few cases in the past (now properly cited) and it requires inventive qualities to combine several fairly distinct processes, resulting in a very complex measurement system. This must definitely be credited to the authors."

Reply: We thank the reviewer for his/her professional and helpful feedback, and appreciate the credit given to our work and our reply letter.

Comment: "They did actually respond satisfactorily to all my questions, except that they could not fully convince me about the reality of a large margin for improvement... That's why I keep convinced that the proposed configuration has little margin for improvement, unless an entirely different scheme is used to retrieve the distributed power of the interacting waves."

Reply: We fully agree with the reviewer that the prospects for signal-to-noise ratio (SNR) and spatial resolution enhancement are central issues. **In the revised manuscript, we report a significant enhancement in SNR without having to increase the number of averaged traces.** The breakthrough was achieved through phase modulation of the light source, at 100 Mbit/s rate. Modulation leads to spectral broadening, which in turn suppresses noise due to coherent Rayleigh backscatter (see also added references). **The higher SNR allows us to report measurements with 50 m resolution over the same length of 3 km.** We could not perform such measurements before. The method and results are presented to the revised manuscript.

This significant progress should provide encouragement for the prospects further improvements in future work. A couple of months ago, we did not consider the particular solution path noted above. We respectfully disagree with the reviewer's certainty in "little margin for improvement".

Comment: "The idea to use a double-pulse technique to improve the spatial resolution is at first glance conceptually correct and makes fully sense. However, the response will be reduced by a proportional amount and here I don't see how they can realistically compensate this attenuated response. If we assume a 10-times better spatial resolution - which is a reasonable target to make the system practical - the noise and fluctuations must be reduced in the same proportion, or the response increased in the same proportion, too. Practically it means to take 100X more wavelength averaging to reduce the coherent Rayleigh noise, which sounds not very realistic. Another option would be to increase the power in the signal by a factor 10 (assuming an additive noise), but the waves will be rapidly depleted by 4-wave mixing, so this is not realistic as well."

Reply: We agree with many of the points raised here. High-resolution analysis would reduce the optomechanical amplification. The weaker gain cannot be simply compensated with higher input power due to the onset of competing nonlinear effects, and increasing the number of averages by an order of magnitude is problematic. However, we disagree with a key issue: the limiting noise mechanism is due to coherent Rayleigh backscatter, which is *not* additive. As noted in the previous reply, we **demonstrate the partial mitigation of this noise through control over the coherence of the light source, with no extra averages.**

Pulse lengths of 500 ns were used in the new reported measurements. Double-pulse analysis would employ longer pulse durations, on the order of several micro-seconds. The application of coherence control to longer pulses would lead to even more effective mitigation of coherent Rayleigh backscatter noise than reported here. We therefore consider the prospects of double-pulse analysis to be quite promising. The discussion of potential double-pulse analysis in the revised manuscript has been modified.

Comment: "I have to say that the proposition to make the 2 optical tones unequal in power to reduce 4-wave mixing is totally incorrect: if P1 is reduced by a factor N and P2 is multiplied by the same factor N, the interacting waves are of unequal power while the response keeps identical since given by the product P1xP2 according to Equ. S20-S21. But it is straightforward to show that the total power summed in the two 4-wave mixing sidebands is minimum when N=1, so when the powers are equal."

Reply: We accept the comment, and thank the reviewer for this correction. This argument is removed from the revised manuscript.

Comment: "I totally follow the authors when they say it is a first step and they make the comparison with the development of distributed Brillouin sensing which reached similarly poor distributed performance in the first publication. We can also counter-argue that we are 30 years later and there was a tremendous improvement in the response of the distributed sensing techniques in the meantime, so today's tools are much better sharpened."

Reply: We agree that 30 years of distributed fiber sensors have provided us with many useful lessons. That being said, only few works to-date addressed the distributed monitoring of forward-scattering processes in fiber (as also noted by the reviewer). This objective remains largely uncharted territory.

Comment: "As I said in my introduction I am fully satisfied by the other responses given by the authors. There are just 2 points on which I need further clarification: 1) In Figure 2, why the amplification of the lower frequency is smaller than the attenuation of the higher frequency, as a result of the interaction? They must exactly compensate according to the equations derived by the authors, but there is nearly a factor 2 which cannot be reasonably credited to experimental artefacts."

Reply: The imbalanced transfer of power seen in Fig. 2 is due to residual four-wave-mixing processes, which transfer part of the input power to additional, higher-order tones. Care is taken to restrict the optical power of these tones to less than 25% of the output power of either of the two primary components. This explanation was added to the revised manuscript.

Comment: "2) In their response the authors clarifies that the gain of their frequency-selective Brillouin amplifier is 5 dB, which is just a factor 3. So it means that there is a non-negligible contribution from the other tone in the recorded trace. This must be mentioned and the impact evaluated."

Reply: We agree with the reviewer that contribution from the unamplified tone towards the output waveform is non-negligible. To work around that potential difficulty, reference traces are taken with the backwards SBS gain window detuned from both components. The reference traces consist of the unamplified contribution of both tones. The subtraction of the reference traces therefore eliminates the residual contribution from unamplified tones. The reviewer is right to note that this procedure should be explained. We added this description to the Methods section of the revised manuscript.

Summary:

All reviewers agree that opto-mechanical time-domain reflectometry represents a novel and interesting breakthrough in fiber sensing and marks a significant achievement. True, the concept has not yet reached the level of performance required for most envisioned applications, and we cannot guarantee that it will. However, as our ongoing advances indicate, such prospects should certainly not be dismissed. **In few months, we already obtained significant progress towards higher resolution, and performed measurements outside long, coated fibers.** The current performance is already respectable enough to bring the idea to the attention of a broad community. The publication of our work in a high-profile journal such as *Nature Communications* would encourage other groups to join the efforts, and help drive this concept to its full potential. We hope that you find this revision of the manuscript suitable for publication.

REVIEWERS' COMMENTS:

Reviewer #1 (Remarks to the Author):

The authors addressed my comments sufficiently and the paper can be published in its current form.